# Activation of the insulin receptor by insulin-like growth factor 2

Weidong An [1,4], Catherine Hall [2,4], Jie Li [1], Albert Hung [2], Jiayi Wu [2], Junhee Park [2], Liwei Wang [1], Xiao-chen Bai [1,3] ✉ & Eunhee Choi [2] ✉

Insulin receptor (IR) controls growth and metabolism. Insulin-like growth factor 2 (IGF2) has different binding properties on two IR isoforms, mimicking insulin's function. However, the molecular mechanism underlying IGF2-induced IR activation remains unclear. Here, we present cryo-EM structures of full-length human long isoform IR (IR-B) in both the inactive and IGF2-bound active states, and short isoform IR (IR-A) in the IGF2-bound active state. Under saturated IGF2 concentrations, both the IR-A and IR-B adopt predominantly asymmetric conformations with two or three IGF2s bound at site-1 and site-2, which differs from that insulin saturated IR forms an exclusively T-shaped symmetric conformation. IGF2 exhibits a relatively weak binding to IR site-2 compared to insulin, making it less potent in promoting full IR activation. Cell-based experiments validated the functional importance of IGF2 binding to two distinct binding sites in optimal IR signaling and trafficking. In the inactive state, the C-terminus of α-CT of IR-B contacts FnIII-2 domain of the same protomer, hindering its threading into the C-loop of IGF2, thus reducing the association rate of IGF2 with IR-B. Collectively, our studies demonstrate the activation mechanism of IR by IGF2 and reveal the molecular basis underlying the different affinity of IGF2 to IR-A and IR-B.

Insulin receptor (IR) is a receptor tyrosine kinase that plays critical roles in cellular signaling, growth, and metabolism[1–10]. IR can be activated by two highly related ligands – insulin and insulin-like growth factor 2 (IGF2)[11,12]. Insulin and IGF2 share high sequence similarity and function but their expression levels vary during embryonic development, between tissues, and under pathophysiological conditions[13–19]. Moreover, it has been known that insulin and IGF2 bind to IR with different affinities and activate the IR in distinct ways, enabling them to execute diverse biological functions through the same receptor in different tissues, metabolic, and developmental conditions.

We and others have demonstrated that multiple insulin molecules binding to two distinct sites of IR (site-1 and site-2) synergistically induce a large conformational change of the IR from an Λ-shaped apo-IR to a Γ-, Τ-, or T-shaped IR dimer, leading to the activation of IR[6,20–24].

Similarly, previous site-directed mutagenesis and binding assays suggest two interaction sites between IR and IGF2[25–29]. However, due to the lack of structural information regarding the IR/IGF2 complex, it is unclear exactly how IGF2 engages site-1 and site-2 of IR. In addition, insulin has two chains (A and B chains) that are linked by three disulfide bonds[30,31], whereas IGF2 is a single-chain molecule with B- and A-domains connected by both disulfide bonds and a C-loop (the C domain including the part of the B domain after the hinge residue)[32,33]. Previous cryo-EM structures reveals that the C-loop of IGF2 is important for the binding of IGF2 to IGF1R[34]; however, whether and how the C-loop of IGF2 is involved in its binding to IR remain unclear.

Additionally, IR has two alternative splicing isoforms (denoted IR-A and IR-B[35,36]), resulting from the skipping or inclusion of exon 11. Consequently, IR-B has an additional 12 amino acids at the C-terminal

[1]Department of Biophysics, University of Texas Southwestern Medical Center, Dallas, TX 75390, USA. [2]Department of Pathology and Cell Biology, Vagelos College of Physicians and Surgeons, Columbia University, New York, NY 10032, USA. [3]Department of Cell Biology, University of Texas Southwestern Medical Center, Dallas, TX 75390, USA. [4]These authors contributed equally: Weidong An, Catherine Hall. ✉e-mail: Xiaochen.Bai@UTSouthwestern.edu; EC3477@cumc.columbia.edu

helix of the alpha subunit (α-CT), while IR-A lacks this segment (Fig. S1a). Insulin binds IR-A and IR-B with a similar affinity[17,37,38]. In contrast, IGF2 has approximately 8 times higher affinity for IR-A than IR-B[37–39]. So far, there is no structural explanation as to why IGF2 binds to IR-B with a lower affinity.

Here, we first solved the cryo-EM structures of IR-A/IGF2 and IR-B/IGF2 complexes at saturated concentrations of IGF2. The conformations of the IR-A/IGF2 complex are identical to those of the IR-B/IGF2 complex, yet both differ from the structure of IR-A resolved under saturated insulin concentrations. Specifically, the structure of IR-A at saturated insulin concentrations exhibits a single T-shaped symmetric conformation[20,21], but both the IR-A and IR-B predominantly adopt T-shaped asymmetric conformations with two or three IGF2 molecules bound at site-1 and site-2. Structural differences of IR induced by insulin and IGF2 are likely due to their different site-2 binding affinities. The site-1 binding of IGF2 is mainly mediated by the interaction between A- and B-domains of IGF2 and L1/α-CT and FnIII-1 domains of IR in a fashion similar to the site-1 insulin binding. In addition, the C-loop of IGF2 also makes weak contacts with the α-CT, CR and L2 domains of IR. Our mutagenesis, binding, and cellular assays, along with IGF2 mutations, demonstrated the functional importance of IGF2-IR interfaces. Furthermore, we showed that Arg30 of IGF2 is a key gatekeeper residue that downregulates the binding affinity of IGF2 to IR site-1.

As the structures of IGF2-bound IR-A and IR-B are identical, it cannot explain why IGF2 binds to IR-B with a lower affinity. To elucidate the molecular basis underlying the distinct binding affinity of IGF2 to IR-A and IR-B, we determined the cryo-EM structure of full-length human IR-B in the apo-state at a resolution of 3.9 Å, showing that, different from the structure of apo-IR-A, the C-terminus of α-CT in the IR-B contacts the FnIII-2 domain of the same IR-B protomer. Such interaction reduces the structural flexibility of IR-B's α-CT and hinders the threading of its C-terminal tail into the C-loop of IGF2, a step known to be the key in the binding of IGF2 to IR. This explains why IGF2 binds to IR-B with a lower affinity than to IR-A. Altogether, our results reveal how IGF2 engages and activates IR, providing insight into the underlying mechanism of ligand-specific IR activation.

## Results

### Overall structure of the IR-A/IGF2 and IR-B/IGF2 complexes

To reveal the mechanism underlying IGF2-dependent IR activation, we determined the cryo-EM structures of full-length human IR-A and IR-B in the presence of saturated IGF2 (Figs.S1–S3). We expressed and purified the full-length of human IR-A and IR-B with a kinase-dead mutation from HEK293F cells (Fig. S1). The cryo-EM structures of IGF2 bound to IR-A and IR-B showed similar conformational landscapes. Particularly, during the initial 3D classification of the IR-A/IGF2 and IR-B/IGF2 particles set, three distinct classes were identified (Figs.S2 and S3). The 3D reconstruction of class 1 (determined at 3.6 Å resolution for IR-A/IGF2, comprising 16% of the particles; 3.7 Å resolution for IR-B/IGF2, comprising 14% of the particles) exhibited a T-shaped symmetric conformation with four IGF2 molecules bound at the canonical site-1 and site-2 (Fig. 1a; Fig. S4a). The symmetric IR-A and IR-B bound with four IGF2 molecules adopts an almost identical conformation to a T-shaped 2:4 IR/insulin complex[20].

The 3D reconstruction of class 2 (determined at 3.8 Å resolution for IR-A/IGF2, comprising 19% of the particles; determined at 4.2 Å resolution for IR-B/IGF2, comprising 22% of the particles) showed an T-shaped asymmetric conformation with three IGF2 molecules bound (Fig. 1b; Fig. S4b). The asymmetric structures of IR-A/IGF2 are nearly identical to that of IR-B/IGF2 (Fig. 1c,d). In one half of the asymmetric IR/IGF2 complex, the two IGF2 molecules bind to site-1' in the head of T-shape and site-2' in the middle regions of the receptor (Fig. 1b). In the other half of the asymmetric complex, the third IGF2, located in the middle region of the T-shaped structure, is sandwiched between site-1

and site-2' from two adjacent IR protomers. This asymmetric IR/IGF2 complex exhibits similar architecture to the recently reported cryo-EM structure of IR/insulin complex at subsaturated insulin concentrations[21]. The 3D reconstruction of class 3 reached lower resolution, for which we were not able to build an atomic model (Fig. 1e). Nevertheless, similar to the class 2, the cryo-EM map of class 3 also displays an asymmetric shape. On one side of the density map, two IGF2 molecules, bound at site-1 and site-2, respectively, are clearly observed. On another side of the density map, the L1/α-CT domains are poorly resolved likely due to the continuous motions.

Interestingly, in the presence of saturated IGF2, most IR-A (84% of total particles) and IR-B (86% of total particles) particles adopt a T-shaped asymmetric conformation, distinct from the IR-A particles in response to saturated insulins showing a single T-shaped symmetric conformation[20,21]. The underlying mechanism that causes such structural differences between IGF2- and insulin-bound IR will be discussed in the following sections.

### IGF2 binding at site-1 of IR

Structural comparison between IR-A/IGF2 and IR-B/IGF2 complexes indicates that the site-1 and site-2 binding modes of IGF2 in IR-A and IR-B are identical. In addition, the 12-residue extension in the C-terminus of α-CT, the unique structural element that only exists in IR-B, was not resolved in the cryo-EM map of IR-B/IGF2 complex, suggesting that this structural element is not involved in the ligand binding or stabilizing the active conformation of IR.

Similar to the site-1 binding of insulin[20,40], IGF2 simultaneously interacts with sites-1a and −1b' between two IR protomers, thereby stabilizing the active conformation of IR (Fig. 2a,b). At site-1a, the A and B domains of IGF2 primarily contact the α-CT' and L1 domains of IR, closely resembling the binding mode of insulin at site-1a[40]. Thus, we will not describe these interactions in detail here. In addition to these interfaces, the C-loop of IGF2 also makes weak contacts with the CR, L2 and α-CT' domains of IR (Fig. 2a). Sidechain densities for most residues in the C-loop of IGF2 were clearly resolved in the cryo-EM map, except for residues 33 − 36, which allowed the precise model building for a major part of the C-loop of IGF2. Our model showed that Arg30 and Ala32 in the N-terminal part of the IGF2's C-loop loosely interacts with a short loop in the CR domain (residues 271-273) of IR. This short loop of the CR domain is not involved in the interaction with insulin, and thus becomes disordered in the structure of IR/insulin complex[20–22]. Two positively charged residues in the C-terminal part of the IGF2's C-loop, Arg37 and Arg38, contact the Tyr708 in α-CT' and are proximal to a cluster of negatively charged residues, including Glu316, Glu318, and Asp322 in L2 domains of IR (Fig. 2a). Similar interaction is also observed in the structure of IGF1R/IGF2 complex[34].

To determine the function of these IGF2-specific IR site-1 interface, we first introduced multiple mutations in IR (E316A, E318A, D322A, and E316A/E318A/D322A) to weaken the interaction between the C-loop of IGF2 and IR L2 domain (Fig. 2c, Fig.S5a). A single mutation in IR L2 domain (E316A, E318A, and D322A) did not affect IGF2-dependent activation. The IR L2 domain triple mutant (E316A/E318A/D322A) exhibited defective IGF2-dependent activation, but still can be fully activated by insulin (Fig. 2c), supporting the notion that the C-loop of IGF2 contributes to the IR/IGF2 interaction by contacting the IR L2 domain.

Besides the primary site-1 binding (site-1a), IGF2 simultaneously touches a secondary sub-site in the site-1 of active IR (site-1b), involving a loop on the top part of the FnIII-1' domain (Fig. 2a). A similar site-1b interaction pattern was observed in the cryo-EM structure of the IR-A/insulin complex[20] (Fig. 2b). However, the residues involved in the site-1b binding are not conserved between insulin and IGF2. In the site-1b of IR/insulin complex, HisB5 of insulin interacts with Pro495 and Phe497 of IR. In contrast, the site-1b interaction in the structure of IR/IGF2 complex is mostly driven by the salt-bridge formed between Glu12 of

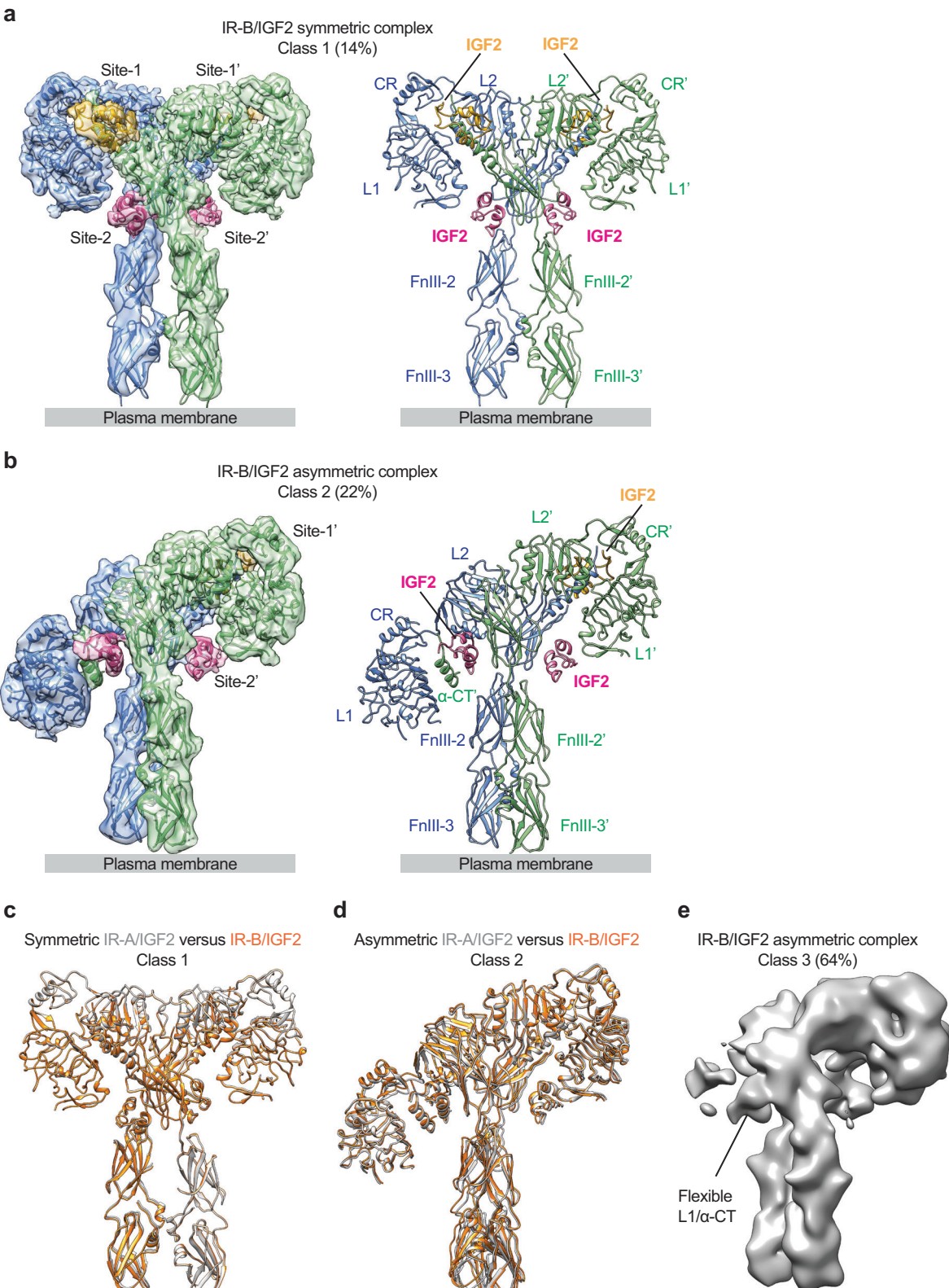

**Fig. 1 | Overall structures of IR-B/IGF2 and IR-A/IGF2 complex. a** 3D reconstruction of IR-B/IGF2 complex in symmetric conformation fitted into a cryo-EM map at 3.7 Å resolution (left). Ribbon representation of the symmetric IR-B/IGF2 complex (right). The two IR protomers are colored in blue and green. IGF2s at site-1 and site-2 are colored in yellow and purple, respectively. **b** 3D reconstruction of IR-B/IGF2 complex in asymmetric conformation fitted into a cryo-EM map at 4.7 Å resolution (left). Ribbon representation of the asymmetric IR-B/IGF2 complex (right). **c** Superposition between IR-B/IGF2 (orange) and IR-A/IGF2 (gray) complex in their symmetric conformation (class 1). **d** Superposition between IR-B/IGF2 (orange) and IR-A/IGF2 (gray) complex in their asymmetric conformation (class 2). **e** Cryo-EM density maps of the IR-B/IGF2 complex in asymmetric conformation (class 3).

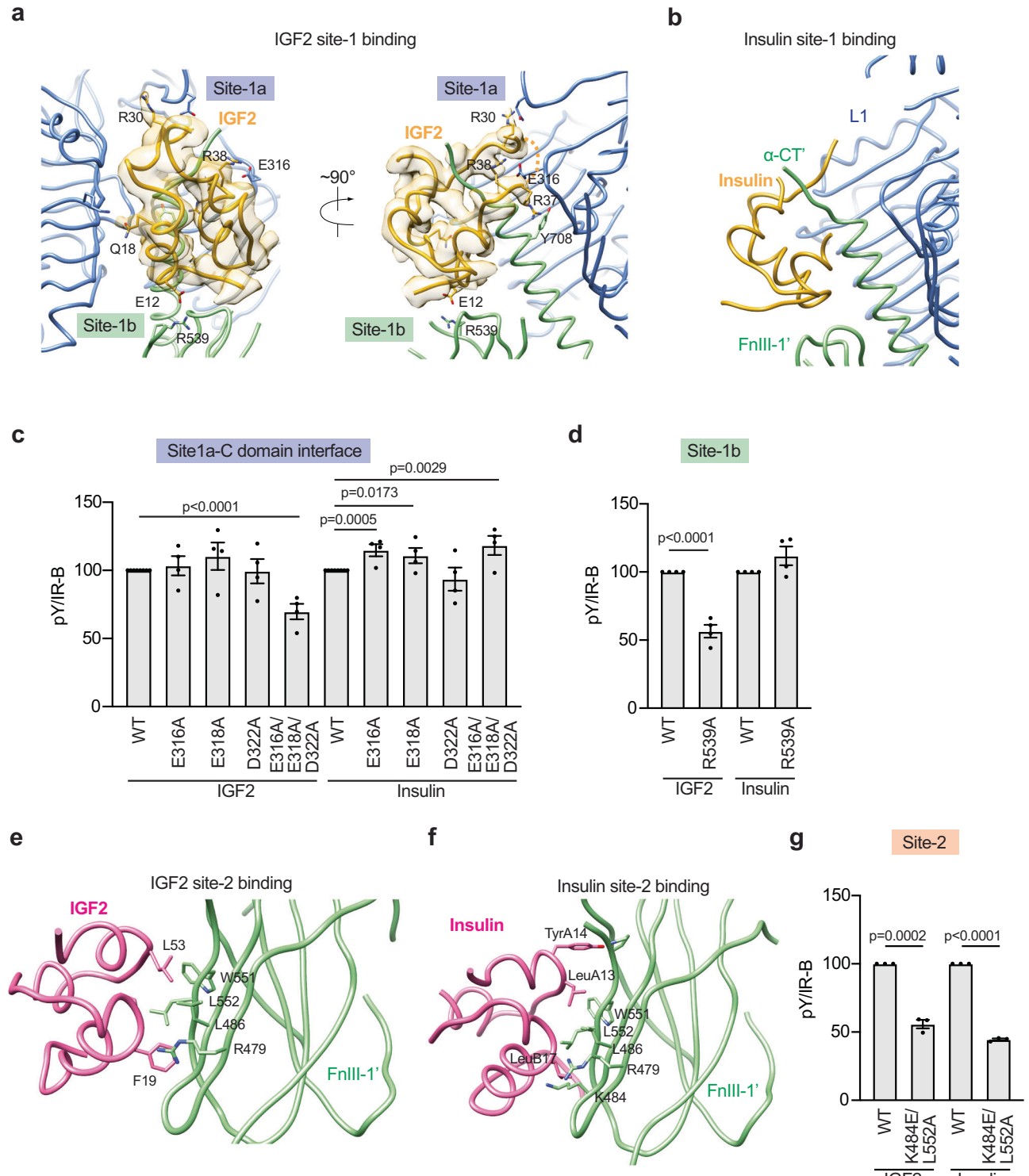

**Fig. 2 | Functional importance of IGF2 binding to two distinct sites. a** Close-up view of the binding of IGF2 (yellow) at the L1 (blue) and α-CT'/FnIII-1' (green) domains of IR. **b** Close-up view of the binding of insulin (yellow) at the L1(blue) and α-CT'/FnIII-1' (green) domains of IR. **c** Autophosphorylation of IR-B by 100 nM IGF2 and 10 nM insulin for 10 min in IR/IGF1R double knockout 293FT cells expressing IR-B WT or the indicated IR-B mutants. Levels of pY IR-B were normalized to total IR-B levels and shown as intensities relative to that of IR-B WT in cells treated with IGF2 (for IGF2-treated group) or insulin (for insulin-treated group). Mean±sem. $N = 4$ independent experiments. Significance calculated using two-tailed Student's *t-test*. Source data are provided as a Source Data file. **d** Autophosphorylation of IR-B by 100 nM IGF2 and 10 nM insulin for 10 min in IR/IGF1R double knockout 293FT cells expressing IR-B WT or IR-B R539A. Levels of pY IR-B were normalized to total IR-B levels and shown as intensities relative to that of IR-B WT in cells treated with IGF2 (for IGF2-treated group) or insulin (for insulin-treated group). Mean±sem. $N = 4$ independent experiments. Significance calculated using two-tailed Student's *t-test*. Source data are provided as a Source Data file. **e** Close-up view of the binding of IGF2 (purple) at the FnIII-1' domain of IR-B (green). **f** Close-up view of the binding of insulin (purple) at the FnIII-1' domain of IR-B (green). **g** Autophosphorylation of IR-B by 100 nM IGF2 and 10 nM insulin for 10 min in IR/IGF1R double knockout 293FT cells expressing IR-B WT or IR-B K484E/L552A. Levels of pY IR-B were normalized to total IR-B levels and shown as intensities relative to that of IR-B WT in cells treated with IGF2 (for IGF2-treated group) or insulin (for insulin-treated group). Mean±sem. $N = 3$ independent experiments. Significance calculated using two-tailed Student's *t-test*. Source data are provided as a Source Data file.

IGF2 and Arg539 of IR. To validate the functional importance of site-1b on IGF2-induced IR activation, we introduced R539A mutation into IR to disrupt the interface at site-1b. Indeed, 293FT cells expressing IR R539A mutant exhibited deficient IGF2-dependent IR activation without affecting insulin-dependent IR activation (Fig. 2d, Fig. S5b).

Altogether, our functional and structural data demonstrated that while IGF2 and insulin bind to IR site-1 in a similar manner, there are two distinct differences. First, in addition to the classical site-1a interfaces between the A and B domains of IGF2 and the IR L1/α-CT domains, the C-loop of IGF2 interacts weakly with the IR α-CT, CR and L2 domains. Second, Glu12 of IGF2 forms a salt bridge with Arg539 in the FnIII-1 domain of IR, which strengths the interaction of IGF2 at IR site-1b.

### IGF2 binding at site-2 of IR

IGF2 binds IR site-2 (the side surface of the FnIII-1 domain of IR) in a similar fashion to that for site-2 insulin binding[5,6,15,20–22], but their detailed binding modes are different (Fig. 2e,f). The buried surface of IGF2 binding at IR FnIII-1 domain (424 Å$^2$) is considerably smaller than that of site-2 insulin binding (681 Å$^2$), indicating that IGF2 engages site-2 with much lower affinity than site-2 insulin binding. Moreover, the residues that are involved in the site-2 binding are not conserved between insulin and IGF2. It has been shown that a total of fourteen insulin residues participate in the site-2 binding[20]. By contrast, only four IGF2's residues, including Phe19, Asp52, Leu53, and Glu57, are involved in the site-2 interaction (Fig. S5c and S6). Previous binding results show that mutations of either Phe19, Leu53, or Glu57 of IGF2 to alanine reduce the binding affinity to IR by approximately two to three times[26], supporting our structural model. To further validate the functional importance of site-2 binding on IGF2-induced IR activation, we introduced double mutations (K484E/L552A) on IR, which is expected to abolish the IR-IGF2 interactions at site-2 (Fig. 2g; Fig. S5d). Consistent with our structural models, IGF2 showed largely reduced potency in activating the IR K484E/L552A mutant, confirming the critical role of site-2 IGF2 binding in IR activation.

The binding of four insulins to both sites-1 and −2 on the IR efficiently promotes the T-shaped symmetric conformation of IR[20,21]. In the asymmetric conformation of IR/IGF2 complex, only two or three IGF2s are bound at sites-1 and −2 (Fig. 1b; Fig. S4). Hence, it is reasonable to speculate that the predominant adoption of asymmetric conformations of IR in response to saturated IGF2 (about 84% and 86% of total particles for IR-A and IR-B, respectively) might result from the comparatively weaker binding of IGF2 at IR site-2. This also partially explains why insulin and IGF2 activate IR with different potencies.

### IGF2 site-1 and site-2 mutants cannot promote optimal IR activation and trafficking

To further investigate the function and mechanism of IGF2 binding on IR activation, we generated IGF2 with a mutation on the residue that is critical for either site-1 or site-2 binding. Specifically, we introduced mutations in IGF2: V43E to disrupt the IGF2/IR α-CT interaction (site-1a); R37A, R38A, and R37A/R38A to disrupt the IGF2/IR L2 interaction (site-1a); E12A to disrupt the IGF2/IR site-1b interaction; F19A/L53A to disrupt the IGF2/IR site-2 interaction; and E12A/R37A/R38A to disrupt IGF2/IR site-1a/b interaction (Figs. 2a, e, and 3a).

To eliminate the effects of IGF2 signaling through IGF1R, we used IR and IGF1R double knockout mouse preadipocytes expressing human IR-B exogenously. We analyzed IR autophosphorylation (pY IR) and the activating phosphorylation of key signaling proteins, including protein kinase B (pAKT) and extracellular signal-regulated kinase 1/2 (pERK) in various concentrations (Fig. 3b,c; Fig.S7). IGF2 V43E (site-1a), E12A (site-1b), and E12A/R37A/R38A (site-1a/b) mutants exhibited greatly decreased potency in triggering IR activation. IGF2/IR L2 interaction defective mutants (IGF2 R37A, R38A, and R37A/R38A) and

IGF2 site-2 mutant (F19A/L53A) could induce robust IR autophosphorylation (Fig. 3b,c; Fig. S7); however, they did not potently activate downstream IR signaling.

The activated IR undergoes clathrin-mediated endocytosis, redistributing and terminating the IR signaling[41–43]. We examined the IGF2-dependent IR endocytosis in HeLa cells stably expressing IR-A WT-GFP (Fig. 3d,e). IGF2 WT induced IR internalization and most IR was found inside cells after 30 min. Although IGF2 R37A/R38A and F19A/L53A promoted IR autophosphorylation (Fig. 3b,c), these mutants were less efficient in inducing IR internalization (Fig. 3d,e), likely due to the defective activation of the MAPK pathway[44]. These results further demonstrate the critical role of both site-1 and site-2 IGF2 binding in promoting optimal IR signaling and trafficking.

### Conformational changes of the C-loop of IGF2 upon IR binding

Superimposing the model of IR-site-1a-bound IGF2 onto the free IGF2 determined previously by NMR revealed an expanding motion of the C-loop of IGF2[32,33] (Fig. 4a,b), which is caused by the threading of the C-terminus of α-CT of IR into the center of the C-loop of IGF2 during the IGF2 binding to the IR site-1a. Since the C-domain of insulin is removed during maturation, such conformational change is not required for insulin to bind to the IR site-1a. This means that insulin pays less entropic and enthalpic costs when binding to IR, which explains its higher affinity to IR as compared to IGF2. Intriguingly, the side chain of Arg30 in free IGF2 is localized in the center of the C-loop, impeding the entrance of the α-CT of IR-B into the C-loop[17,32,33] (Fig. 4b). These structural observations indicate that Arg30 of IGF2 might act as a gatekeeper residue that, to a certain extent, hinders the interaction between IR and IGF2.

To test this hypothesis, we generated IGF2 R30A mutant and examined IGF2 WT and R30A-induced IR activation (Fig. 3a). Strikingly, IGF2 R30A induced the phosphorylation of IR, AKT, and ERK in both IR-A and IR-B expressing cells more potently than IGF2 WT (Figs. 3b, 4c, d; Fig. S5e). We next performed IGF2 binding assay using isolated full-length human IR-A or IR-B (Fig. 4e,f). Consistent with the cell-based signaling assay, IGF2 R30A mutant exhibited approximately three times higher IR-A and IR-B binding affinity compared with IGF2 WT. These results collectively suggest that Arg30 of IGF2 plays role in negatively regulating the IGF2-induced IR activation.

### Cryo-EM structure of long-form IR (IR-B) in the apo-state

As mentioned above, no structural differences were observed in the active structures of IR-A/IGF2 and IR-B/IGF2 complexes, indicating that the different binding affinity of IGF2 to IR-A and IR-B is not related to the active conformation of these two isoforms. To further investigate why IGF2 binds to IR-B with a lower affinity, we determined the cryo-EM structure of the ligand-free, apo-IR-B dimer, at 3.9 Å resolution (Fig. 5a–c; Fig. S8). The overall structure of the IR-B dimer in the apo-state assumes a classical Λ-shape (Fig. 5b,c), similar to the crystal structure of apo-IR-A[45]. The two IR-B protomers tightly pack together primarily through the interaction between L1/L2 domains of one protomer and FnIII-2'/FnIII-1' domains of the other. This Λ-shaped conformation imposes a distance of approximately 120 Å between the two membrane-proximal domains of the IR-B, indicating that this is the inactive form.

Notably, the 12-residue extension in the C-terminus of α-CT of IR-B (Fig. 5a) was not resolved in the cryo-EM map of apo-IR-B, presumably due to structural flexibility. Nevertheless, the residues immediately preceding the 12-residue extension in the C-terminal region of α-CT are well folded and form an extended loop in the structure of apo-IR-B (Fig. 5d–f). As a result, the α-CT motif in the apo-IR-B becomes longer as compared to that in the apo-IR-A, and thus can reach and contact the FnIII-2 domain of the same IR protomer through the interaction between Phe714 from α-CT and Asn594 from FnIII-2 domain (Fig. 5d,f).

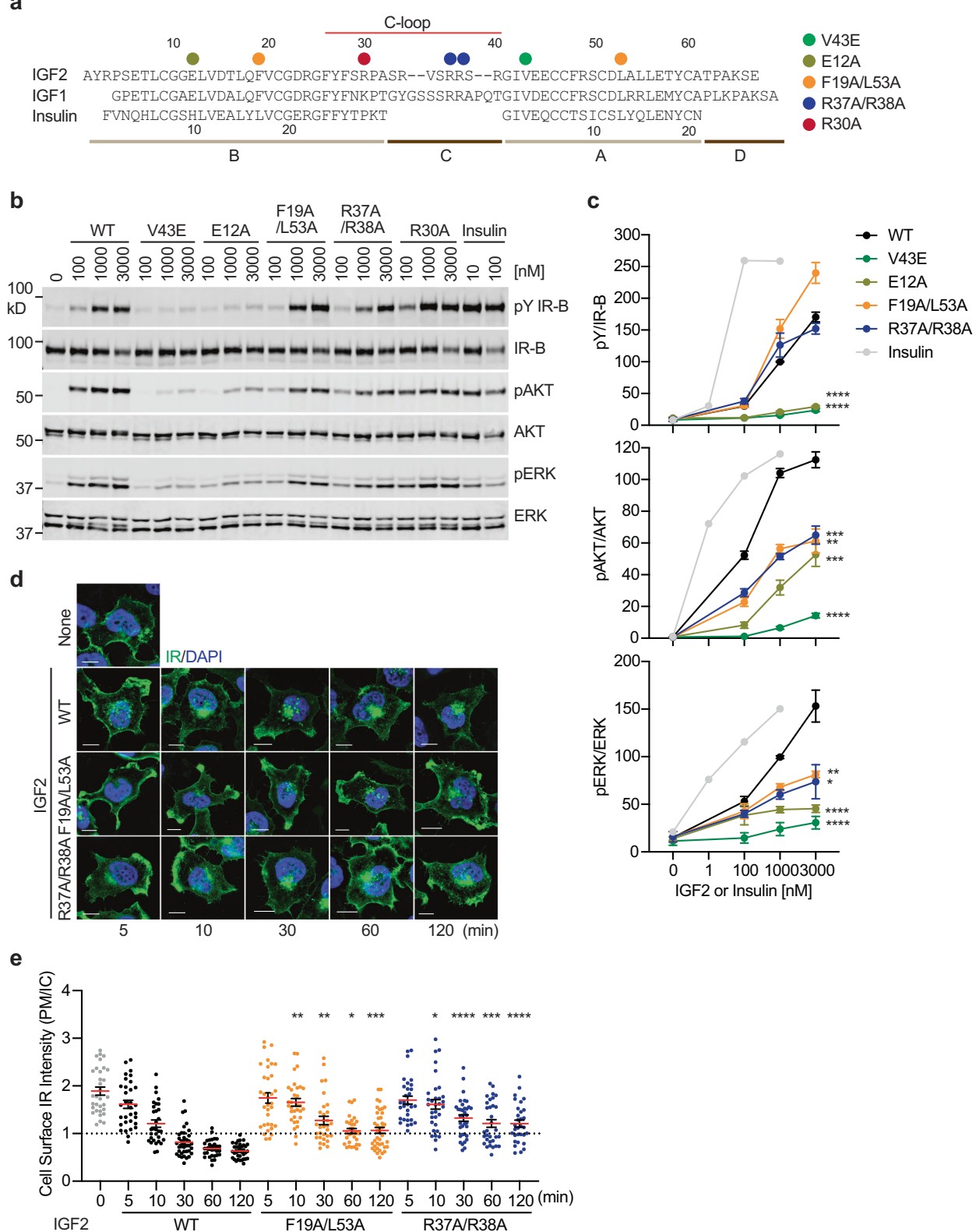

It has been suggested that the binding of IGF2 to IR requires the threading of the C-terminus of α-CT through the C-loop of IGF2[34] (Fig. 4). We, therefore, hypothesized that the additional structural restraint of the C-terminal part of α-CT imposed by the α-CT/FnIII-2 interaction in the apo-IR-B might hinder its threading into the C-loop of IGF2, thus largely reducing the association rate of IGF2 with IR-B. To test this idea, we introduced N594A, N594R, and N594E mutations into IR-B to disrupt the interface. As Phe714 in IR α-CT is crucial for ligand binding, the residue could not be mutated[20]. Indeed, IR-B N594A, N594R, and N594E mutants exhibited increased IGF2- and insulin-dependent IR-B activation, when expressed in 293FT cells (Fig. 5g,h). These data suggest that the structural restraint of the C-terminal segment of α-CT by the α-CT/FnIII-2 interaction regulates ligand-dependent IR-B activation and explain the lower binding affinity of IGF2 to IR-B as compared to IR-A.

**Fig. 3 | Functional importance of IGF2-IR interfaces on IR activation and trafficking. a** Sequences and domains of human insulin, IGF1, and IGF2. Mutations of IGF2 are noted in colors. **b** IR signaling in IR and IGF1R double knockout preadipocytes expressing only human IR-B (DKO-IR-B) treated with the indicated ligands for 10 min. Cell lysates were blotted with the indicated antibodies. Source data are provided as a Source Data file. **c** Quantification of the western blot data shown in (**b**). Levels of protein phosphorylation were normalized to total protein levels and shown as intensities relative to that in 1000 nM IGF2 WT-treated cells. Mean±sem. For pY IR: IGF2 WT, $N = 18$; V43E, $N = 7$; E12A, $N = 6$; F19A/L53A, $N = 5$; R37A/R38A, $N = 5$; insulin, $N = 2$; For pAKT/AKT: WT, $N = 17$; V43E, $N = 6$; E12A, $N = 6$; F19A/L53A, $N = 4$; R37A/R38A, $N = 4$; insulin, $N = 2$; For pERK/ERK: WT, $N = 17$; V43E, $N = 6$; E12A, $N = 6$; F19A/L53A, $N = 4$; R37A/R38A, $N = 4$; insulin, $N = 2$. Significance calculated using two-tailed Student's *t-test*. *$p < 0.05$, **$p < 0.01$, ***$p < 0.001$, and ****$p < 0.0001$. The exact $p$ values are provided in the source data. **d** HeLa cells expressing IR-A WT-GFP were starved, treated with 100 nM IGF2 WT or mutants for indicated times, and stained with anti-GFP (IR, green) and DAPI (blue). Scale bar, 10 μm. **e** Quantification of the ratios of plasma membrane (PM) and intracellular (IC) IR-A-GFP signals of cells in (**d**). Mean±sem. 0 min, $N = 32$; For IGF2 WT: 5 min, $N = 32$; 10 min, $N = 31$; 30 min, $N = 37$; 60 min, $N = 31$; 120 min, $N = 36$; For R37A/R38A: 5 min, $N = 30$; 10 min, $N = 32$; 30 min, $N = 35$; 60 min, $N = 32$; 120 min, $N = 31$; For F19A/L53A, 5 min, $N = 33$; 10 min, $N = 32$; 30 min, $N = 31$; 60 min, $N = 31$; 120 min, $N = 40$. Significance calculated using two-tailed Student's *t-test*; between IGF2 WT and mutants in the same time points. *$p < 0.05$, **$p < 0.01$, ***$p < 0.001$, and ****$p < 0.0001$. The exact $p$ values are provided in the source data.

## IGF2 mimics insulin's function

Although glucose homeostasis is mainly regulated by insulin in the short-term, emerging evidence suggests that IGF2 may also play a significant role in long-term glucose homeostasis[46]. Unlike insulin, which is regulated by meals, IGF2 is abundantly expressed in most tissues throughout life, including metabolic tissues such as the liver and muscle. The IGF2 levels in the blood are 400-1000 ng/ml, approximately 100 times higher than insulin[19]. Therefore, it is tempting to speculate that IGF2 may function in human metabolism through the IR, despite its lower affinity than insulin.

Insulin signaling inhibits hepatic glucose production and release[7,8]. We used in vitro glucose production assay to investigate the metabolic effects of IGF2 WT and mutants (Fig. 6a). Consistent with previous results, dexamethasone and cAMP treatment induced glucose production in primary hepatocytes, while insulin significantly suppressed it[47]. IGF2 WT and R30A also suppressed glucose production in hepatocytes. As expected, the IGF2 site-1a mutant, V43E did not suppress glucose production induced by dexamethasone and cAMP. The IGF2 site-2 mutant, F19A/L53A also did not suppress glucose production, albeit to a lesser extent. Although IGF2 R37A/R38A mutant did not activate IR signaling as effectively as IGF2 WT, it suppressed the glucose production similarly to IGF2 WT. Next, to examine the effects of IGF2 mutants on cell growth and viability, we performed cell growth assay (Fig. 6b). IGF2 V43E, F19A/L53A, and R37A/R38A less potently enhanced cell viability and growth, compared to IGF2 WT. These data validate the importance of IGF2 binding to both IR site-1 and site-2 in IGF2-dependent metabolic and mitogenic functions through the IR.

## Discussion

Insulin and IGF2 have high sequence and structural similarities but they exhibit distinct binding affinities and activities on two IR isoforms. Our structural and functional studies present here uncover the unique binding modes of IGF2 at both site-1 and site-2 of IR with respect to insulin, providing the structural basis for the distinct activation mechanisms of IR by insulin and IGF2.

Our ligand-free IR-B structure shows a classical Λ-shaped conformation like apo-IR-A (Fig. 7a,b). However, the C-terminal region of the α-CT in the apo-IR-B becomes more rigid and contacts the FnIII-2 domain of the same protomer (Fig. 7b). Our functional studies demonstrate that the disruption of the α-CT/FnIII-2 domains interaction increases ligand-dependent IR-B activation. We, therefore, propose that the additional structural restraint of the C-terminal part of α-CT imposed by the α-CT/FnIII-2 interaction might hinder its threading into the C-loop of IGF2, thus largely reducing the association rate of IGF2 with IR-B. This provides a structural explanation for why IGF2 binds to IR-B with lower affinity than IR-A. In sharp contrast, the C-domain of insulin is removed during maturation (Fig. 7c). Consequently, the threading action of α-CT is not required for the binding of insulin to IR, explaining why insulin binds to both IR-A and IR-B with similar affinities. In addition, we show that Arg30 of IGF2 blocks the entrance of the IR α-CT into its C-loop (Fig. 7b). As a result, the binding of IGF2 to IR requires the expanding of the C-loop and the 180° flipping

of the side chain of Arg30 (Fig. 7b). Such additional entropic and enthalpic costs reduce the binding affinity of IGF2 to IR.

At saturated insulin concentrations, four insulin molecules bind to two distinct sites of IR, which promotes the formation of exclusive T-shaped conformation[20,21] (Fig. 7a). Our cryo-EM analysis shows that, even at saturated IGF2 concentrations, only about 16% of IR-A particles and 14% of IR-B particles adopt the symmetric T-shaped conformation with four IGF2 molecules bound, while most particles are trapped in asymmetric conformations with two or three IGF2 molecules bound (Fig. 7b). In the asymmetric state, one IGF2 is simultaneously contacting site-1 and site-2' from two protomers, similar to the IR/insulin complex at subsaturated insulin concentrations[21]. In addition, structural comparisons between IR/insulin and IR/IGF2 complexes suggest that the IR site-2 binding affinity of IGF2 is lower than that of insulin. It has been previously demonstrated that the site-2 insulin binding is essential for the structural transition of IR from asymmetric to symmetric conformation[6,21]. Therefore, it is likely that IGF2's lower efficiency in promoting the T-shaped IR is due to its comparatively weak IR site-2 binding affinity. Altogether, our work provides a putative mechanism underlying the different potency of insulin and IGF2 in activating IR.

It remains unknown what percentage of IR forms symmetric or asymmetric conformations in vivo in response to the binding of insulin or IGF2. Furthermore, it is unclear whether the structural differences of IR (*i.e.*, symmetric or asymmetric conformation) induced by different stoichiometries of bound ligands drive signaling specificity.

Insulin levels vary depending on metabolic condition and tissues. After pulsatile secretion from pancreatic beta cells, insulin is transported rapidly to hepatocytes through the portal vein, where approximately 80% of secreted insulin is cleared by the liver during its first passage through the portal circulation[48]. The half-life of insulin in the portal circulation is approximately 3-5 minutes[49], indicating that local insulin levels in the liver and pancreas will be much higher than circulating insulin levels. In addition, several disease conditions, such as obesity and type 2 diabetes, result in chronic hyperinsulinemia due to an increase in insulin secretion and a decrease in insulin clearance[42,43,50]. It is tempting to speculate that the conformational differences of IR by different stoichiometries of bound insulins may provide a mechanism in fine-tuning the duration and intensity of IR signaling in tissues and during pathogenesis, in response to dynamic changes in insulin before and after meal. Furthermore, IR fully liganded with insulin (*i.e.*, four insulins per IR dimer) could efficiently clear insulin from the blood through receptor-mediated endocytosis, which may prevent hyperinsulinemia and hypoglycemia.

In humans, IGF2 is abundantly expressed in most tissues throughout life. However, levels of functional IGF2 can be controlled by multiple mechanisms, including IGF2R-dependent clearance[51], PAPP-A2 proteolysis[52], and IGF-binding proteins[53]. Therefore, it is difficult to estimate local free IGF2 levels and predict the conformation of IR induced by IGF2 under physiological conditions. We speculate that, similar to insulin-induced IR activation, IR may have different IGF2-binding occupancy and conformations at different ligand levels. Such a

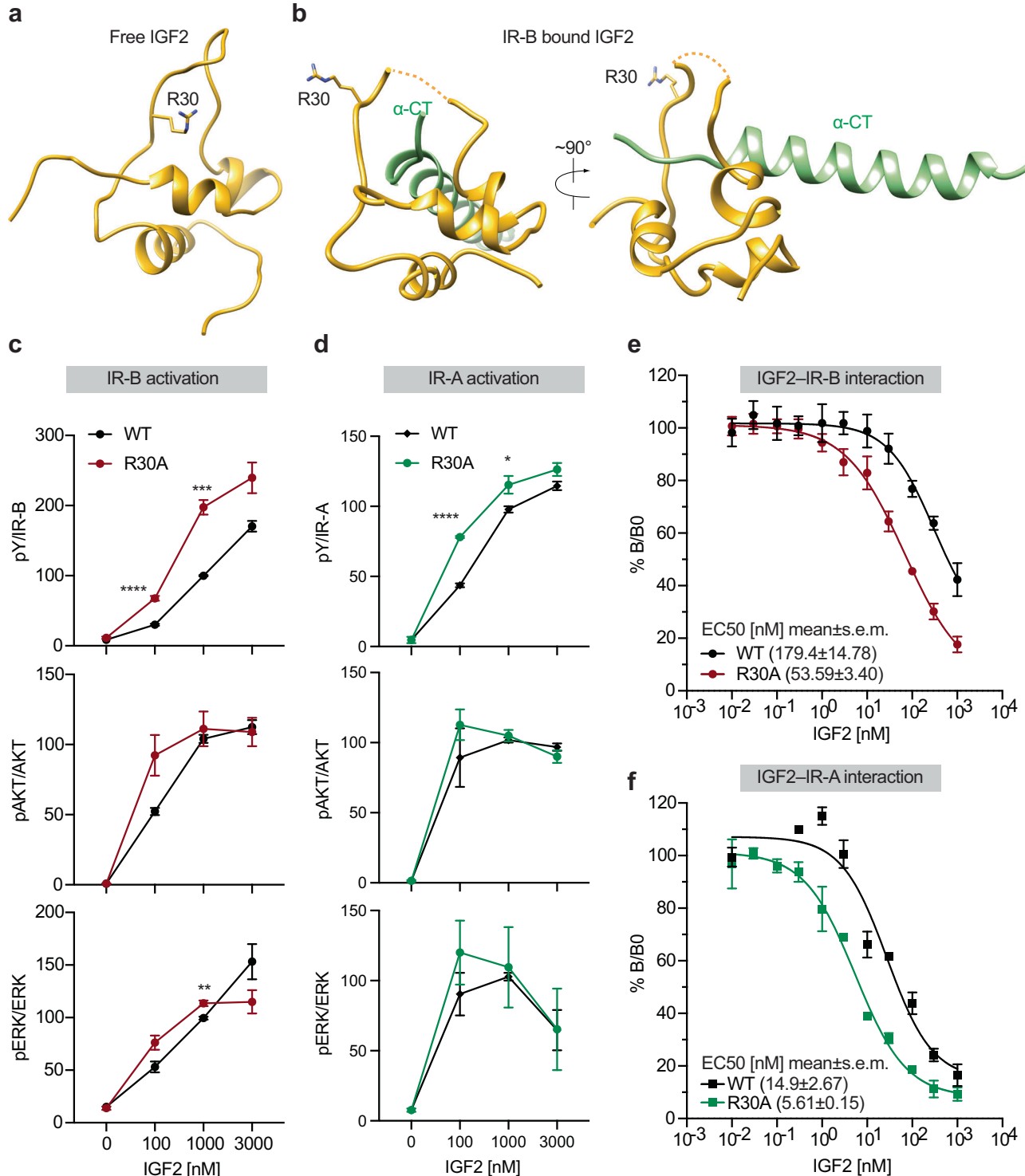

**Fig. 4 | IGF2 R30A mutation enhances IR binding and activation. a** NMR solution structure of IGF2 (PDB:1IGL). **b** Close-up view of the binding of IGF2 (yellow) at the α-CT of IR-B (green). **c** Quantification of the western blot data for IGF2 WT and R30A mutant shown in Fig. 3b. Levels of protein phosphorylation were normalized to total protein levels and shown as intensities relative to that in 1000 nM IGF2 WT-treated cells. Mean±sem. For IGF2 R30A, *N* = 6; For IGF2 WT same as Fig. 3c. Significance calculated using two-tailed Student's *t-test*. **p = 0.0073, ***p = 0.0009, and ****p < 0.0001. Source data are provided as a Source Data file. **d** IR signaling in IR and IGF1R double knockout preadipocytes expressing only mouse IR-A (DKO-IR-

A) treated with the indicated ligands for 10 min. Levels of protein phosphorylation were normalized to total protein levels and shown as intensities relative to that in 1000 nM IGF2 WT-treated cells. *N* = 4 independent experiments. Significance calculated using two-tailed Student's *t-test*; *p = 0.03956 and ****p < 0.0001. Source data are provided as a Source Data file. **e** Competition-binding assay with full-length IR-B. Mean±sd. *N* = 4 independent experiments. EC50 were calculated with best-fit values. Source data are provided as a Source Data file. **f** Competition-binding assay with full-length IR-A. Mean±sd. *N* = 3 independent experiments. EC50 were calculated with best-fit values. Source data are provided as a Source Data file.

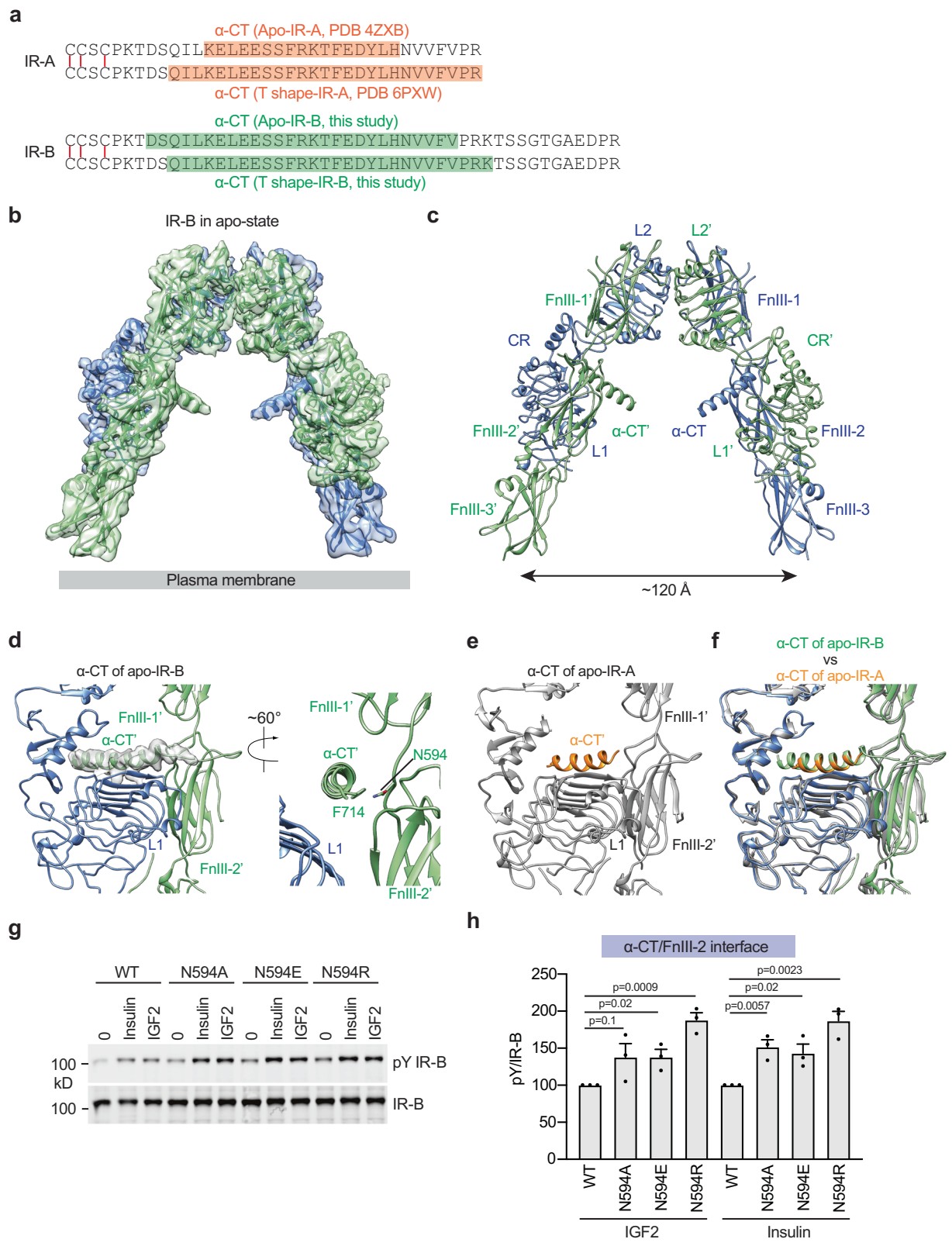

**Fig. 5 | Overall structure of apo-IR-B. a** Sequences of the C-terminal region of α-chains in IR-A and IR-B. Disulfide bonds are marked in red. Structural regions identified in previous and current studies were noted. **b** The 3D reconstruction of IR-B fitted into a cryo-EM map at 3.9 Å resolution. **c** Ribbon representation of IR-B. **d** Close-up view of α-CT of apo-IR-B. **e** Close-up view of α-CT of apo-IR-A (PDB 4ZXB). **f** Superposition between α-CT of apo-IR-B (green) and α-CT of apo-IR-A (orange). **g** Autophosphorylation of IR-B (pY IR-B) by 100 nM IGF2 and 10 nM insulin for 10 min in IR/IGF1R double knockout 293FT cells expressing IR-B wild-type (WT) or the indicated IR-B mutants. **h** Quantification of the western blot data shown in Fig. 5g. Levels of pY IR-B were normalized to total IR-B levels and shown as intensities relative to that of IR-B WT in cells treated with IGF2 (for IGF2-treated group) or insulin (for insulin-treated group). Mean±sem. $N = 3$ independent experiments. Significance calculated using two-tailed Student's *t-test*. Source data are provided as a Source Data file.

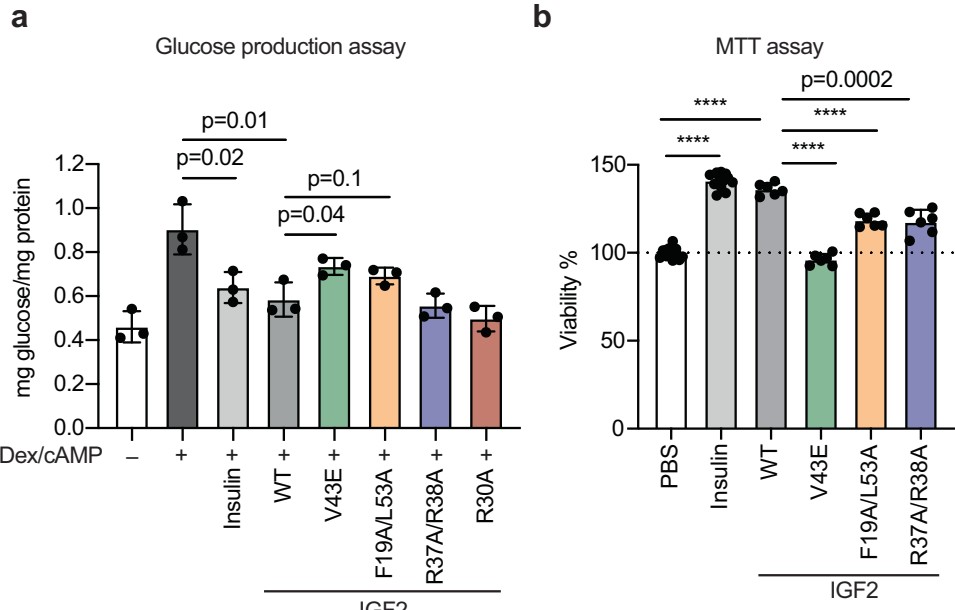

**Fig. 6 | Metabolic and mitogenic effects of IGF2-IR interfaces. a** Function of IGF2 WT and mutants on glucose production inhibition in primary hepatocytes. Mean ±sd. *N* = 3 independent cells. Significance calculated using two-tailed Student's *t-test*. Source data are provided as a Source Data file. **b** Function of IGF2 WT and mutants on cell growth and viability. C2C12 cells were treated with 100 nM ligands for 48 h. The outcomes were presented relative to the cell viability observed in cells treated with PBS. Mean±sd. Control (PBS) and insulin, *N* = 12; IGF2 WT, V43E, R37A/ R38A, and F19A/L53A, *N* = 6. Significance calculated using two-tailed Student's *t-test*; ****$p < 0.0001$. Source data are provided as a Source Data file.

unique multisite system allows IR to respond to a wide range of IGF2 concentrations in various metabolic conditions and tissues.

Since the discovery of IGF2, long-standing questions have focused on how the structural differences of insulin and IGF2 affect their IR binding properties and whether their biological actions are similar or different. Previous studies support the idea that IGF2 activates a bias signaling through the IR-A that may play a distinct role in the growth of fetal tissues and cancer cells[38,54]. However, since the discovery of non-suppressible insulin-like activity in serum at the beginning of the 1960s[55], there has been growing evidence that IGF2 plays a role in metabolism beyond growth control. For instance, tumors secreting IGF2 cause hypoglycemia[56,57], dysregulation of IGF2 levels is closely linked to metabolic risk[58], and IGF2 acts as a metabolic regulator in the interplay between mother and fetus to support fetal growth[59]. Furthermore, IGF2 exerts insulin-like effects in adipocytes by stimulating glucose uptake and inhibiting lipolysis, and in hepatocytes by stimulating glycogen synthesis[60–64]. We showed that IGF2 also potently inhibits glucose production in hepatocytes. These data suggest that both IGF2 and insulin stimulate similar downstream signaling via IR, rather than the bias signaling previously proposed. However, compared to insulin, IGF2 exerts its metabolic action through low-affinity binding to IR. Our functional and structural data demonstrate that IGF2 exhibits a less potent effect on activating the IR than insulin, due to its weaker site-2 affinity. In part, these results explain why IGF2 is a weaker version of insulin.

Moreover, IR is expressed widely in the brain. IGF2 is secreted by the choroid plexus into cerebrospinal fluid (approximately 1000 times more than insulin) and promotes stem cell self-renewal and expansion of neural progenitor cells through IR[65–67]. Therefore, it is tempting to speculate that IGF2 may function in human metabolism and growth through the IR, despite its lower affinity than insulin. As well, nature may have developed a mechanism to reduce the possibility of IGF2-induced hypoglycemia by modulating the binding affinity of IGF2 to IR and by expressing IGF-binding proteins.

Collectively, our structural and functional studies demonstrate a distinctive activation mechanism of IR by IGF2 and provide insight into how different ligands bind the same receptor but can induce diverse intensity of IR signaling. Future studies will be required to determine whether structural differences of the IR induced by insulin or IGF2 contribute to signaling outcomes in different developmental stages and metabolic conditions. Our study presented here serves as an important guide for designing agonist and antagonist specifically targeting on insulin or IGF2-induced IR signaling.

## Methods
### Mice
Animal work described in this manuscript has been approved and conducted under the oversight of the Columbia University Institutional Animal Care and Use Committee. Mice (C57BL/6 J, stock #000664, Jackson laboratory) were fed a standard rodent chow (Lab diet, #5053). All animals were maintained in a specific antigen-free barrier facility (temperature, 68–79 °F; humidity, 30–70%) with 12 h light/dark cycles (6 a.m. on and 6 p.m. off). Two-month-old male mice were used in this study.

### Cell lines
**FreeStyle™ 293-F.** Freestyle™ 293-F cells (Invitrogen, R79007) were cultured in FreeStyle™ 293 Expression Medium and maintained in an orbital shaker in 37 °C incubators with a humidified atmosphere of 5% $CO_2$.

**SF9 cells.** Spdoptera frugiperda (SF9, 11496015) cells were cultured in SF900 II SFM (Gibco) at 27 °C with orbital shaking at 120 rpm.

**293FT.** 293FT (Invitrogen, R70007) were cultured in high-glucose Dulbecco's modified Eagle's medium (DMEM) supplemented with 10% (v/v) fetal bovine serum (FBS), 2 mM L-glutamine, and 1% (v/v) penicillin/streptomycin. Cells were maintained at 37 °C with a humidified atmosphere of 5% CO2. To generate IR and IGF1R double knockout 293FT cells, lentiviruses were packaged in 293FT cells by transfecting the cells with lentiCRISPR vectors (pLenti-human IR-puro (Addgene #98290, gRNA: GGATGAACGCCGGACCTATG); pLenti-human IGF1R-

**a**

At saturated insulin concentrations

Apo-IR-B

IR-B/insulin
Symmetric (100%)

Insulin binding to
site-1 and site-2 →

**b**

At saturated IGF2 concentrations

Apo-IR-B

IR-B/IGF2
Asymmetric (64%)

IR-B/IGF2
Asymmetric (22%)

IR-B/IGF2
Symmetric (14%)

Weak IGF2 binding at site-2 →

Flexible
L1/α-CT

IGF2
R30

push

F1 · F2 · L1'

Extended C-terminus of α-CT contacts FnIII-2, hindering its threading into the C-loop of IGF2

**c**

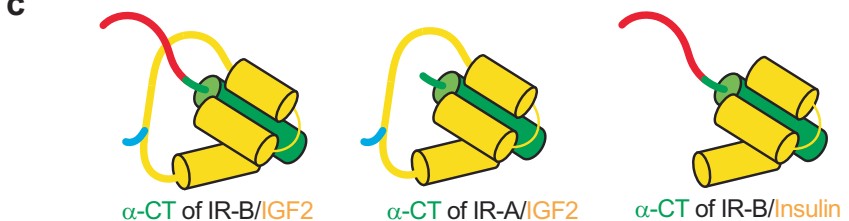

α-CT of IR-B/IGF2       α-CT of IR-A/IGF2       α-CT of IR-B/Insulin

**Fig. 7 | The activation mechanism of IR by IGF2. a** Insulin-induced IR-B activation at saturated insulin concentrations. Two IR-B protomers are colored in blue and green. Insulin at the site-1 and site-2 are colored in yellow and purple, respectively. The C-terminal extension of α-chains in IR-B are marked in red. **b** IGF2-induced IR-B activation at saturated IGF2 concentrations. IGF2 at the site-1 and site-2 are colored in yellow and purple, respectively (top). The C-terminal extension of α-chains contacts the FnIII-2 domain of IR-B, thus hindering its threading into the C-loop of IGF2 (bottom). The side chain of Arg30 of IGF2 is marked in cyan. **c** A comparison of insulin or IGF2-bound α-CT in IR-A and IR-B. Insulin and IGF2 (yellow); α-CT domain (green); the C-terminal extension of α-CT in IR-B (red). The side chain of Arg30 of IGF2 is marked in cyan.

blast (Addgene #52962, gRNA: GGAGAACGACCATATCCGTG)), psPAX2 (Addgene #12260), and pMD2.G (Addgene #12259). Forty-eight and seventy-two hours after transfection, the medium was collected, concentrated with Retro X concentrator (Clonetech), and then added to fresh 293FT cells on with polybrene (4 μg/ml). Cells were subsequently selected with puromycin and hygromycin for two weeks.

**HeLa Tet-on.** HeLa Tet-on cells (Takara Bio) were cultured in high-glucose DMEM supplemented with 10% (v/v) FBS, 2 mM L-glutamine,

and 1 % penicillin/streptomycin and maintained in monolayer culture at 37 °C and 5% CO₂ incubator.

**Brown preadipocytes.** IR and IGF1R double knockout brown preadipocytes expressing only human IR-B or mouse IR-A were kindly provided by Dr. Ronald Kahn[68,69]. Cells were cultured in high-glucose DMEM with 10% FBS, 2 mM L-glutamine, and 1% penicillin/streptomycin. Cells were maintained at 37 °C with a humidified atmosphere of 5% CO₂.

**C2C12.** C2C12 (ATCC, CRL-1772) were cultured in high-glucose DMEM supplemented with 10% (v/v) FBS, 2 mM L-glutamine, and 1% penicillin/ streptomycin and maintained in monolayer culture at 37 °C and 5% CO₂ incubator.

**Cell line validation.** Following passage of an aliquot of each cell line for three to four weeks, a fresh batch of cells was thawed and propagated. There were no signs of mycoplasma contamination.

## Protein expression and purification

**IGF2 expression, refolding, and purification.** DNA sequence encoding IGF2 with the Human Rhinovirus 3 C recognition site followed by a 1 × Flag tag was synthesized by Integrated DNA Technologies (IDT, Inc.), which is cloned into pET-22b (+) with Gibson assembly method to construct pET-22b-IGF2. Plasmids encoding mutant IGF2 ligands were derived from pET-22b-IGF2 with Q5 Site-Directed Mutagenesis Kit (New England Biolabs, Inc.). Sequences of primers are provided in Supplementary Table 1.

To express IGF2 ligands and different mutant derivatives, the plasmids encoding different ligands were transformed into the BL21 bacterial. Transformed *E. coli* BL21 were cultured in 1 L of Lysogeny Broth (LB; 10 g/L tryptone, 5 g/L yeast extract, 10 g/L NaCl) medium containing 100 μg/mL ampicillin at 37 °C. Isopropyl β-d-1-thiogalactopyranoside (IPTG) was added to a final concentration of 0.5 mM when OD600 reaches 1.2-1.4. The cells were cultured at 37 °C overnight. The cells were collected and resuspend with ice-cold lysis buffer (50 mM Tris-Cl, pH 8.0; 50 mM NaCl; 2 mM EDTA-Na) supplemented with Triton-X100 to make a 1% final concentration and PMSF to a final concentration of 0.2 mM. The cell suspension was stirred at 4 °C for 20 min. The cell suspension was sonicated and centrifuged at 4 °C to collect inclusion body. The inclusion body pellet was resuspended with lysis buffer and then added Dithiothreitol (DTT) to the finial concentration of 5 mM and Uracil to the final concentration of 8 M. Then the denatured solution was stirred at room temperature (RT) for 2 h to dissolve the inclusion body. The suspension was further sonicated and centrifuged at 12 °C to obtain supernatant. The supernatant contains denatured IGF2 ligands was slowly dropped into 400 ml refolding buffer (500 mM L-Arginine, pH 8.0; 500 mM NaCl; 1.5 mM Cystine) and stir overnight gently. The refolded solution was centrifuged at 4 °C to precipitate the improper refolded and aggregated proteins. The supernatant containing refolded IGF2 ligands was purified with excel nickel resin. The resin was washed with wash buffer (50 mM Tris-Cl pH 8.0; 400 mM NaCl; 20 mM Imidazole) after binding. The sample was eluted with elution buffer (50 mM Tris-Cl pH 8.0; 400 mM NaCl; 300 mM Imidazole). The samples were concentrated and further purified with Superose 75 increase 10/300 GL size-exclusion column with GF buffer (20 mM HEPES-Na buffer pH 7.5; 400 mM NaCl). The peak fractions correspond to pure IGF2 samples were pooled for structural and cell-based assays.

**Human insulin receptor expression and purification.** For structural studies, plasmid that expresses human insulin receptor of A isoform (hIR-A) and B isoform (hIR-B) were derived from our previously constructed pEZT-BM vector which encodes the protein elements including human Insulin receptor (Y960F, S962A, D1120N, R1333A, I1334A,

L1335A, L1337A), the Human Rhinovirus 3 C recognition site (3 C), affinity purification tag Tsi3 and His8 tag[20]. This plasmid expressing human insulin receptor of A isoform (hIR-A) was inserted with a sequence encoding exon 11 of hIR-B by Q5 Site-Directed Mutagenesis method to construct plasmid expressing hIR-B (pEZT-BM-hIR-B-H3C-Tsi-His8). The primer sequences used to construct hIR-B expression vector were 5′-ACTGGTGCCGAGGACCCTAGGCCATCTCGGAAACGC and 5′- GCCTGA AGAGGTTTTTCTGGGGACGAAAACCACGTTG.

The expression and purification of hIR-A and hIR-B were performed as described in previous studies with some modifications[20,21]. In brief, pEZT-BM-hIR-B-H3C-Tsi-His8 plasmid was transformed into the DH10Bac bacteria to generate bacmid DNA. The bacmid DNA was introduced into Sf9 cells to produce recombinant baculovirus by using Cellfectin reagent. Protein was expressed in FreeStyle 293-F cells by infecting the virus at a ratio 1:10 (virus: cell, v/v). The infected cells were supplemented with 10 mM sodium butyrate to boost protein expression. The cells were cultured for 40-50 h at 30 °C and 8% CO2.

The cells were resuspended in lysis buffer (40 mM Tris–HCl pH 8.0; 400 mM NaCl and Protease Inhibitor Cocktail). The membrane fraction was obtained by ultracentrifugation of the cell lysate for 2 h at 100,000 g at 4 °C. Dodecyl maltoside (DDM) was added to a final concentration of 1.2% with stirring to extract the hIR-A and hIR-B from the membrane fraction at 4 °C overnight. The supernatant containing solubilized protein was obtained by ultracentrifugation for 1 h at 100,000 g at 4 °C. The supernatant was added with 2 mM CaCl2 and Tse3 protein conjugated Sepharose resin and incubate for 1-2 h. The supernatant together with resin was loaded into a column. The resin was washed with wash buffer 1(40 mM Tris–HCl pH 8.0, 400 mM NaCl, 2 mM CaCl2, 0.08% DDM) by gravity flow for 2 times and then washed with wash buffer 2 (40 mM Tris–HCl pH 8.0; 400 mM NaCl, 2 mM CaCl2, 0.05% DDM) for 2 times. hIR-A and hIR-B were eluted by HRV-3C protease cleavage with wash buffer 2. The protein was then concentrated and run on a Superose 6 increase 10/300 GL size-exclusion column with GF buffer (20 mM HEPES-Na buffer pH 7.5; 400 mM NaCl and 0.03% DDM). The dimer fraction was pooled and refolded IGF2 ligands were added at a 2:8 (m/m) ratio. After incubation for 2 h, the protein was concentrated to 6-8 mg/ml for cryo-EM analyses.

## EM data acquisition

EM data acquisition, image processing, and model building and refinement were performed following previous protocols with minor modifications[21,22,70,71].

The cryo-EM grid was prepared by applying 3 μl of the IR-A and IR-B samples (6 mg/ml) in the absence or presence of IGF2 to glow-discharged Quantifoil R1.2/1.3 300-mesh gold holey carbon grids (Quantifoil, Micro Tools GmbH, Germany). Grids were blotted for 4.0 s under 100% humidity at 4 °C before being plunged into the liquid ethane using a Mark IV Vitrobot (FEI). Micrographs were acquired on a Titan Krios microscope (FEI) operated at 300 kV with a K3 direct electron detector (Gatan), using a slit width of 20 eV on a Bio-Quantum energy filter. SerialEM 3.8 was used for the data collection[72]. A calibrated magnification of 46,296 was used for imaging of the samples at both low and high pH, yielding a pixel size of 1.08 Å on images. The defocus range was set from −1.6 μm to −2.6 μm. Each micrograph was dose-fractionated to 30 frames with a total dose of about 60 e-/Å².

## Image processing

The cryo-EM refinement statists for both apo and IGF2 bound IR-B datasets is summarized in Table 1. 4,395 movie frames of IR-B were motion-corrected and binned two-fold, resulting in a pixel size of 1.08 Å, and dose-weighted using MotionCor2.1[73]. The CTF parameters were estimated using Gctf[74]. RELION4 was used for the following processing[75]. Particles were first roughly picked by using the Laplacian-of-Gaussian blob method, and then subjected to 2D classification. Class averages representing projections of IR-B in different orientations were

**Table 1 | Cryo-EM Data Collection and Refinement Statistics**

| | Apo-IR-B (C2) 8U4B EMD-41877 | IR-B/IGF2 (Symmetric) 8U4C EMD-41878 | IR-B/IGF2 (Asymmetric) 8U4E EMD-41880 | IR-A/IGF2 (Symmetric) 8VJB EMD-43279 | IR-A/IGF2 (Asymmetric) 8VJC EMD-43280 |
|---|---|---|---|---|---|
| **Data collection and processing** | | | | | |
| Magnification | 46,296 | 46,296 | 46,296 | 46,296 | 46,296 |
| Voltage (kV) | 300 | 300 | 300 | 300 | 300 |
| Electron exposure ($e^-/Å^2$) | 60 | 60 | 60 | 60 | 60 |
| Defocus range (μm) | 1.6–2.6 | 1.6–2.6 | 1.6–2.6 | 1.6–2.6 | 1.6–2.6 |
| Pixel size (Å) | 1.08 | 1.08 | 1.08 | 1.08 | 1.08 |
| Symmetry imposed | C2 | C2 | C1 | C1 | C1 |
| Initial particle images (no.) | 1,519,317 | 3,962,294 | 3,962,294 | 3,422,813 | 3,422,813 |
| Final particle images (no.) | 53,302 | 74,973 | 119,411 | 66,937 | 79,813 |
| Map resolution (Å) | 3.9 | 3.6 | 4.2 | 3.6 | 3.8 |
| FSC threshold | 0.143 | 0.143 | 0.143 | 0.143 | 0.143 |
| **Refinement** | | | | | |
| Initial model used (PDB code) | 4ZXB | 6PXV | 6PXV | 6PXV | 6PXV |
| Model composition | | | | | |
| Nonhydrogen atoms | 13,196 | 14,700 | 14,175 | 14,688 | 13,910 |
| Protein residues | 1634 | 1826 | 1723 | 1824 | 1724 |
| Ligands | | | | | |
| R.m.s. deviations | | | | | |
| Bond lengths (Å) | 0.002 | 0.004 | 0.003 | 0.002 | 0.003 |
| Bond angles (°) | 0.641 | 0.681 | 0.522 | 0.593 | 0.659 |
| **Validation** | | | | | |
| MolProbity score | 2.05 | 2.12 | 2.84 | 2.09 | 2.18 |
| Clashscore | 10.41 | 14.35 | 73.78 | 13.39 | 15.25 |
| Poor rotamers (%) | 0.27 | 0.00 | 0.00 | 0.18 | 0.26 |
| Ramachandran plot | | | | | |
| Favored (%) | 91.28 | 92.80 | 91.44 | 92.74 | 91.79 |
| Allowed (%) | 8.59 | 6.86 | 8.32 | 7.04 | 8.04 |
| Disallowed (%) | 0.12 | 0.34 | 0.24 | 0.23 | 0.18 |

used as templates for reference-based particle picking. Extracted particles were binned three times and subjected to 2D classification. Particles from the classes with fine structural feature were selected for 3D classification using an initial model generated from a subset of the particles in RELION4. Particles from one of the resulting 3D classes showing good secondary structural features were selected and re-extracted into the original pixel size of 1.08 Å. Subsequently, we performed finer 3D classification with C2 symmetry imposed by using local search in combination with small angular sampling, resulting two new good class with improved density for the entire complex. The cryo-EM map after 3D refinement was resolved at 4 Å resolution.

4366 movie frames of IR-B with IGF2 bound were motion-corrected and binned two-fold, resulting in a pixel size of 1.08 Å, and dose-weighted using MotionCor2. CTF correction was performed using GCTF1.06. Particles were first roughly picked by using the Laplacian-of-Gaussian blob method, and then subjected to 2D classification. Class averages representing projections of the IGF2 bound IR-B in different orientations were used as templates for reference-based particle picking. A total of 3,962,294 particles were picked from 4,366 micrographs. Particles were extracted and binned by three times (leading to 3.24 Å/pixel) and subjected to another round of 2D classification. Particles in good 2D classes were chosen (1,801,808 in total) for 3D classification using an initial model generated from a subset of the particles in RELION4. We performed 3D classification of the selected particles into six classes using a local search in combination

with small angular sampling, resulting in three different types of conformational states. One class shows perfect symmetry, one class show asymmetric conformations with both L1/α-CT domains well resolved, and the other 4 classes also show asymmetric conformation but with one of the L1/α-CT domains poorly resolved. The final reconstructions of the symmetric complex and the rigid asymmetric complex were resolved at 3.7 and 4.7 Å resolution, respectively. The resolution of site-1 bound IGF2 and the surrounding domains were improved to 3.5 Å resolution after focused refinement.

4199 movie frames of IR-A with IGF2 bound were collected. A total of 3,422,813 particles were picked and extracted. After 2D classification, 1,621,860 were selected for 3D classification. We performed 3D classification of the selected particles into six classes using a local search in combination with small angular sampling, resulting in three different types of conformational states. Similar to the IR-B/IGF2 datasets, one class shows perfect symmetry, one class show asymmetric conformations with both L1/α-CT domains well resolved, and the other 4 classes also show asymmetric conformation but with one of the L1/α-CT domains poorly resolved. The final reconstructions of the symmetric complex and the rigid asymmetric complex were resolved at 3.6 and 3.8 Å resolution, respectively.

**Model building and refinement**
Model buildings of IR-B in apo and IGF2 bound states, and IR-A in IGF2 bound states were initiated by rigid-body docking of individual

domains from the structures of L1, CR, L2, and FnIII1-3 domains of IR-A[20]. Manual building was carried out using the program Coot0.8.8[76]. The model was refined by using the real-space refinement module in the Phenix package (V1.17)[77]. Restraints on secondary structure, backbone Ramachandran angels, residue sidechain rotamers were used during the refinement to improve the geometry of the model. MolProbity 4.5 as a part of the Phenix validation tools was used for model validation (Table 1). Figures were generated in Chimera 1.17[78].

## Insulin receptor signaling assay

The IR signaling assay were performed as described earlier with some modification[20–22,71,79,80]. For IR mutant activation assay, the long isoform of human IR (hIR-B) in pCS2-Myc resistant to IR gRNAs was used. The plasmid expressing the short isoform of human IR (hIR-A) was inserted with a sequence encoding exon 11 of hIR-B by Q5 site-directed mutagenesis to generate hIR-B (pCS2-hIR-B-Myc, primer sequences: 5′- and tggtgccgaggaccctagGCCATCTCGGAAACGCAG and 5′- gtgcctgaagaggttttttCTGGGGACGAAAACCACG). To generate gRNA resistant construct, the gRNA target sequence in pCS2-hIR-B-Myc was mutated by Q5 site-directed mutagenesis (primer sequence: 5′- aggaggACCTATGGGGCCAAGAGT and 5′- ctcgtcCGAAAAGGTGACCAGGGTC). Sequences of primers are provided in Supplementary Table 1.

Plasmid transfection was performed with Lipofectamine 2000 (Invitrogen) to express Myc-tagged IR mutants into IR/IGF1R double knockout 293FT cells. After 1 day, the cells were serum starved for 16 h. Serum-starved cells were treated for 10 min with human insulin (I2643, Sigma) diluted in high-glucose DMEM without serum to a concentration of 10 nM or human IGF2 (I2526, Sigma) diluted to 100 nM.

For IGF2 mutant activation assay, IR/IGF1R double knockout preadipocytes expressing human IR-B or mouse IR-A were used[68,69]. Two days after seeding, the cells were serum starved for 6 h. Serum-starved cells were treated for 10 min with homemade IGF2-WT or IGF2 mutants diluted to concentrations of 100, 1000, or 3000 nM in high-glucose DMEM without serum, or human insulin (I2643, Sigma) diluted to 10 or 100 nM.

After treatment, cells were incubated with cell lysis buffer B [50 mM Hepes pH 7.4, 150 mM NaCl, 10% (v/v) Glycerol, 1% (v/v) Triton X-100, 1 mM EDTA, 10 mM sodium fluoride, 2 mM sodium orthovanadate, 10 mM sodium pyrophosphate, 0.5 mM dithiothreitol (DTT), 2 mM phenylmethylsulfonyl fluoride (PMSF)] supplemented with cOmplete Protease Inhibitor Cocktail (Roche) and PhosSTOP (Roche) on ice for 1 h. After centrifugation at 18,213 g at 4 °C for 20 min, cell lysate samples were made with 6X SDS buffer. Cell lysates were analyzed by SDS-PAGE and Western blotting. Anti-IR-pY1150/1151 (1:2000, 19H7, Cell signaling; labeled as pY IR, Cat. #3024), anti-Myc (1:2000; 9E10, Roche; labeled as IR, Cat. #11667149001), anti-IR (1:500; CT3, Santa Cruz, Cat. #sc-57342), anti-AKT (WB, 1:2000; 40D4, Cat. #2920), anti-pS473 AKT (WB, 1:2000; D9E, Cat. #4060), anti-ERK1/2 (WB, 1:2000; L34F12, Cat. #4696), and anti-pERK1/2 (WB, 1:2000; 197G2, Cat. #4377) were used as primary antibodies. For quantitative Western blots, anti-rabbit immunoglobulin G (IgG) (H + L) (Dylight 800 conjugates, Cat. #5151) and anti-mouse IgG (H + L) (Dylight 680 conjugates, Cat. #5470) (Cell signaling) were used as secondary antibodies. The membranes were scanned with the Odyssey Infrared Imaging System (LI-COR, Lincoln, NE). For IR mutant activation assay, levels of pY-IR were normalized to total IR levels and shown as intensities relative to that of IR-WT treated with insulin. For IGF2 mutant activation assay, levels of pY-IR were normalized to total IR levels and shown as intensities relative to the condition of 1000 nM IGF2-WT treatment.

## Immunofluorescence assay

The IR trafficking assay were performed as described earlier with some modifications[22,81]. HeLa cells expressing IR-GFP were plated on coverslips, starved for 14 h, and then treated with 100 nM IGF2 for the indicated times. Cells were fixed with cold methanol for 10 min, washed with 0.1% Triton X-100 in PBS (0.1% PBST) and incubated with 3% bovine serum albumin (BSA) in 0.1% PBST for 1 h. The cells were incubated with diluted anti-GFP antibodies (1:500, homemade[81]) in 3% BSA in 0.1% PBST overnight at 4 °C. After wash, the cells were incubated with fluorescent secondary antibodies (1:200, Alexa fluor 488 goat anti-rabbit IgG (H + L) antibody, Cat. #A11008, Invitrogen) and mounted on microscope slides in ProLong Gold Antifade reagent with DAPI (Invitrogen). Images were acquired with a Leica thunder Imager (Leica Microsystems Inc.). The cell edges were defined with Image J. The whole cell (WC) signal intensity and intracellular (IC) signal intensity were measured. The plasma membrane (PM) signal intensity was calculated by subtracting IC from WC. Identical exposure times and magnifications were used for all comparative analyses.

## In vitro IGF2-binding assay

In vitro IGF2-binding assay was conducted as previously described with slight modification[21,22]. To isolate human IR-A and IR-B, 293FT cells were transfected with pCS2-IR-A-Myc or pCS2-IR-B-Myc using Lipofectamin™ 2000 (Invitrogen). Three days later, the cells were serum starved for 14 h. The cells were lysed with cell lysis buffer B without Dithiothreitol (DTT) supplemented with cOmplete™ Protease Inhibitor cocktail (Roche) and PhosSTOP (Sigma) on ice for 1 h. After centrifugation at 20,817 g at 4 °C for 10 min, the concentrations of cell lysate were measured using Micro BCA Protein Assay Kit (Thermo Fisher Scientific). Cell lysates and anti-c-Myc magnetic beads (Cat. #88842, Thermo Fisher Scientific, 250 ug of beads per 3 mg of total cell lysates) were incubated at 4 °C for 2 h. The beads were washed two times with the washing buffer B [50 mM Hepes pH 7.4, 400 mM NaCl, 0.05% NP-40] supplemented with cOmplete™ Protease Inhibitor cocktail (Roche) and PhosSTOP (Sigma). The beads were washed once with binding buffer [20 mM Hepes pH 7.4, 200 mM NaCl, 0.03% Dodecyl maltoside (DDM), and 0.003% cholesteryl hemisuccinate (CHS) (Anatrace)] supplemented with cOmplete™ Protease Inhibitor cocktail (Roche) and 100 nM BSA and resuspended in the binding buffer. 10 ul of IR-bound beads, 1 nM Alexa Fluor® labeled human insulin, and the indicated amount of homemade IGF2 were incubated on a rotator at 4 °C for 14 h. The beads were washed once with the binding buffer. The bound proteins were eluted with 50 µl of binding buffer containing 2% SDS at 50 °C for 10 min. The samples were diluted with 150 µl of binding buffer. The fluorescence intensities were measured in a microplate reader (Cytation 5; Biotek). Non-specific binding was measured in samples of Alexa Fluor® labeled insulin analogs with beads without IR and subtracted from the data.

## Glucose production assay

Primary hepatocytes were isolated from 2-month-old male mice (C57BL/6Jackson #000664) with a standard two-step collagenase perfusion procedure as described earlier with some modifications[21,71,81]. Briefly, following anesthesia, the inferior vena cava was cannulated, and the liver was perfused with Liver Perfusion Medium (Thermo Scientific, Cat. #17701038) using peristaltic pump (perfusion rate as 3 ml/min). After 1-2 sec upon appearance of white spots in the liver, we cut the portal vein with scissors to wash out blood, and then the liver was perfused with 30 ml of Liver Digest Medium (Thermo Scientific, Cat. #17703034). Dissected liver was gently washed with low glucose DMEM and transferred to sterile culture dish containing 15 ml Liver Digest Medium. The isolated liver cells were filtered through the 70 µm cell strainer into a 50 ml tube. After centrifuge at 50 g for 5 min at 4 °C, cells were washed with cold low glucose DMEM three times. Cells were resuspended with attached medium [Williams' Medium E supplemented with 5% (v/v) FBS, 10 nM insulin, 10 nM dexamethasone, and 1% penicillin/streptomycin]. A million isolated primary hepatocytes were seeded in 35 mm collagen (Sigma, Cat. #C3867)-coated dishes. After 2 h, the medium was

changed to low-glucose DMEM medium supplemented with 10% (v/v) FBS and 1% penicillin/streptomycin for 24 h.

Glucose production inhibition assay was performed as described earlier with some modifications[47]. Briefly, hepatocytes were then serum starved with low-glucose DMEM supplemented with 1% penicillin/streptomycin overnight. After washing hepatocytes twice with PBS, the medium was replaced with glucose production medium (DMEM without glucose, glutamine, and phenol red (Thermo Scientific, Cat.# A1443001) supplemented with 20 mM calcium lactate, 2 mM sodium pyruvate, 1% (w/v) fatty-acid free BSA, and 1% penicillin/streptomycin). 1 mL glucose production medium was added to each cell culture dish along with following additions: (1) vehicle, (2) 100 uM cAMP and 1 uM dexamethasone (Dex/cAMP), (3) Dex/cAMP and 10 nM human insulin (I2643, Sigma), (4-8) Dex/cAMP and 1000 nM of homemade WT IGF2 or IGF2 mutants. After 6 h of incubation, medium was collected, and glucose concentration was measured using Glucose (GO) Assay Kit (Sigma, Cat. #GAGO20) and subsequently normalized to protein content.

## MTT assay
Two thousand C2C12 cells were seeded per well in 96-well plates. In the following day, the cells were serum starved for 4 h and treated with 100 nM ligands for 48 h. After incubation, 20 μl of MTT solution (5 mg/mL in PBS, Sigma, Cat. #M5655) was added to each well. The cells were incubated at 37 °C for another 2 h. After removing the medium, the cells were lysed using 200 μl of DMSO for 30 min at 37 °C. Absorbance was determined at 570 nm using a microplate reader.

## Statistical analysis
Prism 10 was used for the generation of graphs and for statistical analyses. Results are presented as mean ± sd or mean ± sem. Two-tailed unpaired $t$ test were used for pairwise significance analysis. Two-way ANOVA followed by the Dunnett test was used for multiple comparisons. Power analysis for sample sizes were not performed. Randomization and blinding methods were not used, and data were analyzed after the completion of all data collection in each experiment.

## Reporting summary
Further information on research design is available in the Nature Portfolio Reporting Summary linked to this article.

## Data availability
All reagents generated in this study are available with a completed Materials Transfer Agreement. All cryo-EM maps and models reported in this work has been deposited into EMDB/PDB database, under the entry ID: EMD-41877 (Apo-IR-B), 8U4B (Apo-IR-B), EMD-41878 (Symmetric IR-B/IGF2), 8U4C (Symmetric IR-B/IGF2), EMD-41880 (Asymmetric IR-B/IGF2), 8U4E (Asymmetric IR-B/IGF2), EMD-43279 (Symmetric IR-A/IGF2), 8VJB (Symmetric IR-A/IGF2), EMD-43280 (Asymmetric IR-A/IGF2), and 8VJC (Asymmetric IR-A/IGF2). PDB used in this study are as follows: 1IGL, 4ZXB and 6PXV]. Source data are provided with this paper.

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

## Acknowledgements

Cryo-EM data were collected at the University of Texas Southwestern Medical Center (UTSW) Cryo-Electron Microscopy Facility, funded in part by the Cancer Prevention and Research Institute of Texas (CPRIT) Core Facility Support Award RP170644. We thank Dr. Stoddard for facility access, Drs. Ronald Kahn and Hirofumi Nagao for sharing IR/IGF1R double knockout preadipocytes, and Dr. Youshin Suh for sharing the plate reader. This work is supported in part by grants from the National Institutes Health (R35GM142937 and R01DK132361 to E.C.; R01GM136976 to X.-C.B.), Columbia Diabetes Center (P30DK063608 to E.C.), Columbia Digestive and Liver Diseases Research Center (1P30DK132710 to E.C.), the Welch foundation (I-1944 to X.-C.B.), the Irma T. Hirschl award (to E.C.), and the Alice Bohmfalk Charitable (to E.C.). X.-C.B. is Virginia Murchison Linthicum Scholar in Medical Research at UTSW. The content in this manuscript is solely the responsibility of the authors and does not necessarily represent the official views of the NIH.

## Author contributions

X.-C.B. and E.C. designed and supervised research; all the authors performed research (W.A. and J.L. prepared samples for Cryo-EM; C.H. and J.W. performed cellular experiments; A.H. performed glucose production assay; J.P. performed MTT assay; W.A. and L.W. prepared IGF2 mutants) and analyzed data; X.-C.B. and E.C. wrote the paper with the input from other authors.

## Competing interests

The authors declare no competing interests.
