## [Peer Review File · Nature Communications]

Reviewers' Comments:

Reviewer #1:

Remarks to the Author:

This manuscript is a modified version of a paper submitted to NSMB. Three cryo-EM structures are presented; IGF2 in complex with two isoforms of IR, IR-A and IR-B, and an unliganded structure of IR-B, which is compared to the previously determined unliganded IR-A structure.

IGF2 binds to IR with lower affinity than insulin and stimulates different biological outcomes. Further, whereas insulin binds with a similar affinity to both isoforms, IGF2 binds with an 8-fold higher affinity to IR-A than IR-B. The data for IGF2 in complex with IR-A is new in this resubmitted manuscript, but the two complexes are very similar and do not suggest a reason for the difference in affinity. The unliganded IR-B structure does provide some clues as to the lower affinity for IGF2 compared to IR-A.

In terms of potential relevance for IR signaling, the IR conformations induced by saturating IGF2 are different from saturating insulin. IGF2 induced mostly asymmetric complexes of both IR-A and IR-B with about a quarter of the particles showing 3 IGF2s bound and a more stable complex, whereas the remaining ~ 3/4 have only 2 IGF2s bound and show substantial disorder in the L1/alphaCT region on one side of the molecule. A small fraction of the particles (19-22%) adopt a symmetric complex with 4 IGF2s bound.

The paper does not set up any biological question to address. The cellular data are presented as confirmation of the structural observations only and do not providing major new insights into IGF2 biology.

Overall, the work seems more suited for a more structurally focused journal and does not provide insight that will be of interest to the wider audience of *Nature Communications*.

This manuscript is rather a difficult read and rather poorly written with multiple typos and grammar mistakes.

Additional specific points:

Line 76-77: "... with two or three IGF2 molecules bound at site-1 and site-2, respectively."

This sentence is quite confusing and the term "respectively" seems inappropriate.

The description of the IGF2 binding sites is not very clear. For example, in Lines 117-120, the description is quite hard to follow. It would be beneficial if some additional cues to orient the reader to which IGF2 molecule is being described as this paragraph proceeds.

Figure 3 data mixes IR-B (3b,c) with IR-A (3d,e) – it is not indicated why or whether the results would be the same if the alternate IR form is used.

Label class 1 and class 2 in extended data Figures 2 and 3

Reviewer #2:

Remarks to the Author:

The authors have responded appropriately and in detail to the concerns of the prior reviews. The study has been markedly strengthened by inclusion of additional data and cryo-EM structures. The

Discussion has been sharpened to indicate boundaries of our present knowledge in a thoughtful way. I recommend publication in its present form.

Our point-by-point responses are listed below. For ease of reading, we have colored our responses in blue.

Reviewer #1 (Remarks to the Author):

This manuscript is a modified version of a paper submitted to NSMB. Three cryo-EM structures are presented; IGF2 in complex with two isoforms of IR, IR-A and IR-B, and an unliganded structure of IR-B, which is compared to the previously determined unliganded IR-A structure.

IGF2 binds to IR with lower affinity than insulin and stimulates different biological outcomes. Further, whereas insulin binds with a similar affinity to both isoforms, IGF2 binds with an 8-fold higher affinity to IR-A than IR-B. The data for IGF2 in complex with IR-A is new in this resubmitted manuscript, but the two complexes are very similar and do not suggest a reason for the difference in affinity. The unliganded IR-B structure does provide some clues as to the lower affinity for IGF2 compared to IR-A.

In terms of potential relevance for IR signaling, the IR conformations induced by saturating IGF2 are different from saturating insulin. IGF2 induced mostly asymmetric complexes of both IR-A and IR-B with about a quarter of the particles showing 3 IGF2s bound and a more stable complex, whereas the remaining ~ 3/4 have only 2 IGF2s bound and show substantial disorder in the L1/alphaCT region on one side of the molecule. A small fraction of the particles (19-22%) adopt a symmetric complex with 4 IGF2s bound. The paper does not set up any biological question to address. The cellular data are presented as confirmation of the structural observations only and do not provide major new insights into IGF2 biology. Overall, the work seems more suited for a more structurally focused journal and does not provide insight that will be of interest to the wider audience of Nature Communications. This manuscript is rather a difficult read and rather poorly written with multiple typos and grammar mistakes.

Response: Dysfunctional IR signaling causes multiple diseases including diabetes and cancer. In humans, IR can be activated by insulin and IGF2. In contrast to insulin, which is produced in the pancreas and its production is regulated by glucose levels, IGF2 is ubiquitously expressed throughout the body and is one of the most abundant hormones in the blood. Extensive studies have shown that these two structurally related ligands bind the IR with different potencies and may activate the IR in different ways. During the past 20 years, we and others have demonstrated the insulin-dependent IR activation; however, it remains unclear how IGF2 activates the IR and executes diverse biological functions through the same receptor. Additionally, IGF2 exhibits different binding potencies to two splicing isoforms of IR (IR-A and IR-B), whereas insulin binds them with a similar affinity. As most of structural studies have focused on IR-A, the molecular basis of ligand specificity of two IR isoforms is unclear.

In the current study, we determined the first cryo-EM structures of IR-B in both its apo- and IGF2-bound active states, and IR-A in IGF2-bound active states. Combining cell biology and biochemistry approaches, we demonstrated that the interaction between the C-terminus of alpha-CT and the FnIII-2 domain of IR-B reduces the flexibility of the alpha-CT motif that is required to bind to IGF2. This explains why IGF2 binds to IR-B with a relatively low affinity. More intriguingly, our cryo-EM structure of IR/IGF2 complex reveals that IGF2 predominantly induces asymmetric IR conformations, distinct from insulin, which exclusively promotes a symmetric IR conformation. These conformational differences of IR are mainly due to the different site-2 affinity of insulin and IGF2.

Altogether, our work reveals for the first time the mechanism underlying the IGF2-induced IR activation and paves the way for understanding how insulin and IGF2 control pleiotropic biological

functions through activating the same receptor. Our studies raise questions regarding whether structural differences in the IR induced by insulin or IGF2 have a significant impact on signaling outcomes in different developmental stages and metabolic conditions. Further, our study presented here serves as an important guide for designing agonist and antagonist specifically targeting on insulin- or IGF2-induced IR signaling. Therefore, we believe it is suitable for the publication in *Nature Communications*.

Additional specific points:

Line 76-77: "... with two or three IGF2 molecules bound at site-1 and site-2, respectively." This sentence is quite confusing and the term "respectively" seems inappropriate.

Response: Point accepted. We have revised the manuscript.

The description of the IGF2 binding sites is not very clear. For example, in Lines 117-120, the description is quite hard to follow. It would be beneficial if some additional cues to orient the reader to which IGF2 molecule is being described as this paragraph proceeds.

Response: Point accepted. We have revised the manuscript and figure.

Figure 3 data mixes IR-B (3b,c) with IR-A (3d,e) – it is not indicated why or whether the results would be the same if the alternate IR form is used.

Response: Our structural analysis of IR-A/IGF2 and IR-B/IGF2 complexes has revealed that IGF2 is bound to both IR-A and IR-B in the same manner. We do not expect significant differences in the results when an alternative IR isoform is used.

Label class 1 and class 2 in extended data Figures 2 and 3

Response: We have revised Extended data Figures 2 and 3.

Reviewer #2 (Remarks to the Author):

The authors have responded appropriately and in detail to the concerns of the prior reviews. The study has been markedly strengthened by inclusion of additional data and cryo-EM structures. The Discussion has been sharpened to indicate boundaries of our present knowledge in a thoughtful way. I recommend publication in its present form.

Response: We thank the reviewer's positive assessment of our manuscript.